

# Implementation of implicit filter for spatial spectra extraction

Kacper Nowak[1], Sergey Danilov[1,2], Vasco Müller[1], and Caili Liu[1,3]

[1]Alfred Wegener Institute, Helmholtz Centre for Polar and Marine Research, Bremerhaven, Germany
[2]Constructor University, Bremen, Germany
[3]Ocean University of China, Qingdao, China

**Correspondence:** Kacper Nowak (kacper.nowak@awi.de)

**Abstract.** Scale analysis based on coarse-graining has been proposed recently as an alternative to Fourier analysis. It is now broadly used to analyze energy spectra and energy transfers in eddy-resolving ocean simulations. However, for data from unstructured-mesh models it requires interpolation to a regular grid. We present a high-performance Python implementation of an alternative coarse-graining method which relies on implicit filters using discrete Laplacians. This method can work on arbitrary (structured or unstructured) meshes and is applicable to the direct output of unstructured-mesh ocean circulation atmosphere models. The computation is split into two phases: preparation and solving. The first one is specific only to the mesh. This allows for auxiliary arrays that are then computed to be reused, significantly reducing the computation time. The second part consists of sparse matrix algebra and solving linear system. Our implementation is accelerated by GPUs to achieve unmatched performance and scalability. This results in processing data based on meshes with more than 10M surface vertices in a matter of seconds. As an illustration, the method is applied to compute spatial spectra of ocean currents from high-resolution FESOM2 simulations.

## 1 Introduction

Motions in the atmosphere and ocean are characterized by a multitude of spatial scales. Which scales contribute most to the kinetic and available potential energy, energy generation and dissipation over scales, and how the energy is transferred between different scales, such as for example, the scales of gyres and those of mesoscale and submesoscale motions in oceanic dynamics, are among questions frequently asked. The prevailing concept is that of an energy spectrum or cross spectrum.

The most common technique to extract a spatial spectrum is the Fourier transform. However, when working with the output of Ocean General Circulation Models (OGCM) direct application of the Fourier transform is rarely possible as it requires data (samples) to be equally spaced as well as the domain to be in a rectangular shape (in global atmospheric configurations spherical harmonics are generally used). New convolution (coarsening)-based approaches to this problem have been proposed (Aluie et al. (2018); Sadek and Aluie (2018); Aluie (2019)) and there already are multiple practical contributions showing the utility and significance of the proposed approach (see, e.g. Schubert et al. (2020); Rai et al. (2021); Storer et al. (2023); Buzzicotti et al. (2023)). While they solve some of the problems, like domain shape, they require data to be on a regular longitude-latitude grid. For vector fields in spherical geometry the procedure requires preliminary calculation of the Helmholtz decomposition.





Several recent OGCMs such as MPAS-Ocean (Ringler et al. (2013)), FESOM2 (Danilov et al. (2017)) and ICON-o (Korn (2017)) are based either on unstructured triangular meshes, or their dual, quasi-hexagonal meshes. The use of the aforementioned coarse-graining method for the output of such models would require interpolation of the output data from native unstructured mesh to a regular mesh. This means additional computations. More importantly, the horizontal divergence of the interpolated velocities may show marked differences compared to the divergence on the original meshes.

These issues can be avoided if coarse-graining is done on the original meshes. Recently a method has been proposed by Danilov et al. (2023) that solves this task. The method is using implicit filters based on discrete Laplacians. The discrete Laplacian operators can be constructed for arbitrary meshes and data placement on these meshes. This method can therefore work on any mesh and can be applied directly to the output of unstructured-mesh models.

In this paper a high performance Python implementation of the implicit filter method is presented and practical examples of its usage are given. We use simulations performed with FESOM2 to illustrate the performance of the method. The discrete Laplacians depend on the mesh and data placement. For convenience, in section 2 we recapitulate some mathematical details of the method and discretizations. The remaining part of this section discusses implementation. The results obtained with the implicit filters are compared with those produced by convolution based methods using velocity fields from simulation performed with FESOM2 on a global mesh with the resolution of 1 km in the Arctic Ocean in sections 3 and 4. The performance overview of the implementation is presented in the following section 5.

## 2 Implicit filter

### 2.1 Mathematical introduction

Let $\phi(\mathbf{x})$ be a scalar field, with $\mathbf{x}$ lying in some domain $D$. The goal is to find the distribution of the second moment of this field over spatial scales. This can be achieved using a coarse-graining, akin to the methods presented by Aluie et al. (2018) and Sadek and Aluie (2018). However, coarse-graining will rely on implicit filters, as proposed by Danilov et al. (2023). The coarsened field $\overline{\phi}_\ell(\mathbf{x})$ is found by solving:

$$(1 + \gamma(-\ell^2\Delta)^n)\overline{\phi}_\ell = \phi, \tag{1}$$

where $\Delta$ is the Laplacian, the smoothing scale is parameterized by $\ell$, and $\gamma$ is a parameter that tunes the relation of $1/\ell$ to wavenumbers. Below we will take $\gamma = 1/2$. As explained in Danilov et al. (2023), in this case $k_\ell = 1/\ell$ has the sense of wavenumber, and the wavelength is $\lambda = 2\pi\ell$. Such $\ell$ is related to the scale of box filter $\ell_{\text{box}}$ approximately as $\ell_{\text{box}}/\ell = 3.5$. The integer $n$ defines the order of the implicit filter, as discussed in Guedot et al. (2015). The implicit coarsening procedure can be applied to a vector field $\mathbf{u}$ as

$$(1 + \gamma(-\ell^2\Delta)^n)\overline{\mathbf{u}}_\ell = \mathbf{u}. \tag{2}$$





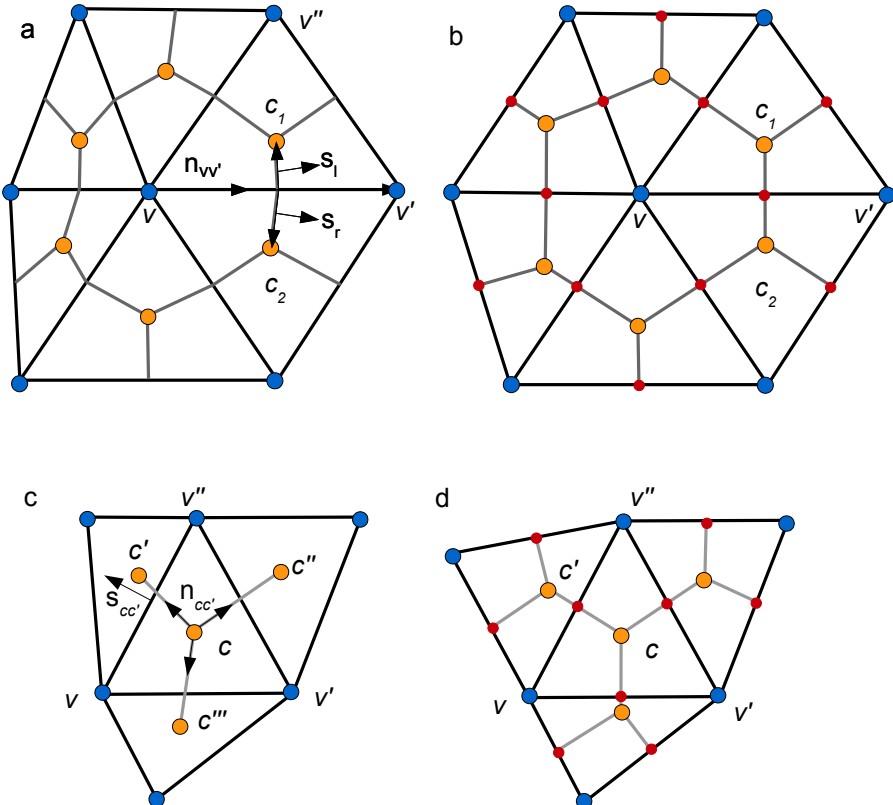

**Figure 1.** Schematics of several unstructured-grid discretizations: (a) Median-dual control volumes around vertices are obtained by connecting centroids with mid-edge points (gray lines); (b) A dual quasi-hexagonal mesh is obtained by connecting the circumcenters of triangles (on orthogonal meshes). The gray lines are perpendicular to edges; (c) Centroids of triangles are used; (d) Circumcenters of triangles are used.

In this case $\Delta$ is the vector Laplacian, which includes metric terms in spherical geometry. In both cases of scalar and vector fields, equations (1) and (2) are complemented by the boundary conditions of no normal flux (for $n = 1$) and additional conditions of no higher-order normal fluxes for $n > 1$.

The discrete Laplacian operator, used in this coarsening method, can be formulated for any computational mesh, whether it is structured or unstructured, or mesh geometry, whether it is flat or spherical. For unstructured meshes, Laplacians can be

discretized through finite volume or finite element methods. Although mathematical details and discretizations were presented in Danilov et al. (2023), we briefly overview them here for convenience. Since discretizations of Laplacians on structured meshes are generally known, we focus below in this section on unstructured meshes.

Figure 1 presents schematics of several unstructured-grid discretizations in 2D view. In FESOM2, scalar degrees of freedom are placed at vertices, and median-dual control volumes are used. They are obtained by connecting centroids of triangles with



mid-edge points, as shown in panel (a). The discrete horizontal velocities are placed on centroids of triangles (panel (c)). The placement of scalars in MPAS-Ocean differs by using the control volumes obtained by connecting circumcenters of triangles (panel (b)). These control volumes are Voronoi quasi-hexagonal polygons of the dual mesh (and vice versa, a triangular mesh can be considered as dual of the hexagonal one). The vector degrees of freedom are in this case the components of velocity normal to the edges of the scalar cells. In ICON-o, scalar degrees of freedom are placed at the circumcenters of triangles, and

normal velocities are at mid-edges (panel (d)). The discretization of Laplacians depend on the placement of the degrees of freedom.

### 2.1.1   Scalar Laplacians

For median-dual control volumes, we use the finite element method, assuming first $n = 1$. The weak formulation of (1) is obtained by multiplying (1) by a sufficiently smooth function $w(\mathbf{x})$ and integrating over the domain $D$. This leads to

$$\int_D (w\overline{\phi}_\ell + (\ell^2/2)\nabla w \cdot \nabla \overline{\phi}_\ell)dS = \int_D w\phi dS.$$

The boundary term appearing after in the integration is zero by virtue of the boundary conditions. The discrete fields are expanded in series $\overline{\phi}_\ell = \sum_{v'} \overline{\phi}_{v'} N_{v'}(\mathbf{x})$ and $\phi = \sum_{v'} \phi_{v'} N_{v'}(\mathbf{x})$, where $N_v(\mathbf{x})$ is a $P_1$ piecewise linear basis function. This function equals 1 at the position of vertex $v$ (see Fig. 1), decays to 0 at vertices connected to $v$ by edges (like $v'$ and $v''$), and is zero outside the stencil of nearest triangles. The continuous Galerkin approximation is obtained by requiring the weak equation above be valid for $w = N_v$ for any $v$. This results in

$$(\mathsf{M}_{vv'} + (1/2)\tilde{\mathsf{D}}_{vv'})\overline{\phi}_{v'} = \mathsf{M}_{vv'}\phi_{v'}.$$

Here, the summation over repeating $v'$ is implied; $\mathsf{M}_{vv'} = \int N_v N_{v'} dS$ is the mass matrix and $\tilde{\mathsf{D}}_{vv'} = \int \ell^2 \nabla N_v \cdot \nabla N_{v'} dS$. Keeping the full mass matrix in this case does not improve the accuracy and it can be replaced by its diagonally lumped

approximation $\mathsf{M}^L_{vv'} = A_v \mathsf{I}_{vv'}$, where $A_v$ is the area of control volume associated with $v$. Note that the system matrix $\mathsf{S} = \mathsf{M}^L + (1/2)\tilde{\mathsf{D}}$ is symmetric and positive definite.

In the biharmonic case, $\mathsf{S} = \mathsf{M}^L + (1/2)\tilde{\mathsf{D}}(\mathsf{M}^L)^{-1}\tilde{\mathsf{D}}$. The derivation procedure consists of two steps. One writes $\Delta\Delta\overline{\phi}_\ell = \Delta\overline{\psi}_\ell$, with $\overline{\psi}_\ell = \Delta\overline{\phi}_\ell$. Using the weak form of the last equality and expanding $\overline{\psi}_\ell$ in the same set of basis functions one gets $\mathsf{M}^L_{vv'}\overline{\psi}_{v'} = \tilde{\mathsf{D}}_{vv'}\overline{\phi}_{v'}$. On the second step one applies the finite element method to $\overline{\phi}_\ell + (1/2)\Delta\overline{\psi}_\ell = \phi$. The flux terms that would

appear in the weak equations are omitted by virtue of boundary conditions. The procedure can be generalized to higher-order filters.

The procedure above can be given a finite volume treatment. Turning to Fig. 1 (a), we first compute $\nabla\overline{\phi}_\ell$ on triangles using three vertex values ($v, v'$ and $v''$ for triangle $c_1$) and then combine fluxes through the segments of boundary (for edge $vv'$ there are two segments with area vectors $\mathbf{s}_l$ and $\mathbf{s}_r$ taken with gradients at $c_1$ and $c_2$ respectively. Such treatment will lead to the

same result.





For the quasi-hexagonal control volumes the $-\ell^2\Delta$ operator is expressed in a finite volume way as

$$A_v(\mathsf{D}\overline{\phi}_\ell)v = \ell^2 \sum_{v' \in N(v)} \frac{\overline{\phi}_v - \overline{\phi}_{v'}}{d_{vv'}} l_{vv'}, \tag{3}$$

where $d_{vv'}$ is the length of edge $vv'$ and $l_{vv'}$ is the distance between circumcenters $c_1$ and $c_2$ on both sides of edge $vv'$ (see Fig. 1 (b)). The system matrix is

$$\mathsf{S}_{vv'}\overline{\phi}_{v'} = A_v\phi_v, \quad \mathsf{S}_{vv'} = A_v\mathsf{I}_{vv'} + (1/2)A_v\mathsf{D}_{vv'} \tag{4}$$

with $\mathsf{I}_{vv'}$ being the identity matrix. The summation over repeating index $v'$ is implied. On uniform meshes, $(\mathsf{M}^L)^{-1}\tilde{\mathsf{D}} = \mathsf{D}$. The matrix of the biharmonic operator can be obtained by applying the procedure used for D twice,

$$\mathsf{S}_{vv''} = A_v\mathsf{I}_{vv''} + (1/2)A_v\mathsf{D}_{vv'}\mathsf{D}_{v'v''}. \tag{5}$$

For scalars at triangle circumcenters, the expression is similar to (3)

$$A_c(\mathsf{D}\overline{\phi}_\ell)c = \ell^2 \sum_{c' \in N(c)} \frac{\overline{\phi}_c - \overline{\phi}_{c'}}{d_{cc'}} l_{cc'}, \tag{6}$$

where $A_c$ is the area of triangle $c$, $N(c)$ is the set of neighboring triangles, $l_{vv'}$ is the length of edge $vv'$ and $d_{vv'}$ is the distance between the circumcenters of triangles with common edge $vv'$ (see Fig. 1 (d)). The biharmonic case is similar to the previous case. In the last two cases the expressions for Laplacians are simplified because of the orthogonality of the lines connecting centroids to edges.

### 2.1.2 Vector Laplacians

For the discretization of FESOM2 the easiest method is to seek for $\overline{\mathbf{u}}_\ell$ at vertices even though the discrete $\mathbf{u}$ are at centroids. One reason is that the number of vertices is twice smaller, leading to matrix problem of smaller dimension. Due to the appearance of metric terms, equations for both components of $\overline{\mathbf{u}}_\ell$ are coupled, as explained in Danilov et al. (2023). The resulting matrix problem is $\mathsf{S}_2 = \mathsf{M}_2 + (1/2)\mathsf{D}_2$ for the Laplacian filter, and $\mathsf{S}_2 = \mathsf{M}_2 + (1/2)\mathsf{D}_2(\mathsf{M}_2^L)^{-1}\mathsf{D}_2$ for the biharmonic filter. It acts on the vector $(\{\overline{u}_v\}, \{\overline{v}_v\})^T$. Here

$$\mathsf{D}_2 = \begin{pmatrix} \tilde{\mathsf{D}} & \mathsf{T} \\ -\mathsf{T} & \tilde{\mathsf{D}} \end{pmatrix}, \quad \mathsf{M}_2 = \begin{pmatrix} \mathsf{M} & 0 \\ 0 & \mathsf{M} \end{pmatrix}.$$

The entries of matrix $\mathsf{T}$ are computed as $\mathsf{T}_{vv'} = \ell^2 \int m(-N_{v'}\partial_x N_v + N_v\partial_x N_{v'})dS$. It is the operator accounting for metric terms linear in the metric factor $m = R_e^{-1}\tan\theta$, where $R_e$ is Earth radius, which is taken constant on triangles, and $\theta$ is the latitude. Compared to the scalar case, the entries of $\tilde{\mathsf{D}}$ are also modified by the metric terms as $\tilde{\mathsf{D}}_{vv'} = \ell^2 \int (\nabla N_v \cdot \nabla N_{v'} + (R_e^{-2} + m^2)N_vN_{v'})dS$. Finally, the right hand side is obtained by projecting from cell to vertices: $\mathbf{u}_v = \mathsf{R}_{vc}\mathbf{u}_c$, where $\mathsf{R}_{vc} = \int N_vM_cdS$, and $M_c$ is the indicator function on triangle $c$. Summation is implied over repeating indices in matrix-vector products.



There are several options to do the filtering keeping $\overline{\mathbf{u}}_\ell$ at cells. For the stencil of nearest triangles (see Fig. 1 (c)) the lines connecting the centroids on general meshes are not perpendicular to edges. For this stencil, there is no universally valid discretization for Laplacian. We use a simplified expression instead of true $-\ell^2\Delta$

$$A_c(\mathbf{Du})_c = A\ell^2 \sum_{c'\in C(c)} (\mathbf{u}_c - \mathbf{u}_{c'}) \tag{7}$$

which works stable in practice. One gets a valid discretization for $-\ell^2\Delta$ taking $A = 1$ on uniform quadrilateral meshes and $A = \sqrt{3}$ on uniform triangular meshes composed of equilateral triangles. On a triangular mesh obtained by splitting regular squares it corresponds to $-\partial_{xx} - \partial_{yy} \pm \partial_{xy}$ for $A = 3/2$, with the sign dependent on the direction the quadrilateral cells are split. The appearance of mixed derivatives is caused by the low symmetry of the mesh (only to rotations by 180 degrees). Such meshes will make scale analysis essentially anisotropic, however all operators on such meshes have a similar mesh imprint. The operator (7) is symmetric and positive semidefinite. In spherical geometry, the unit zonal and meridional vectors are different at $c$ and $c'$ locations. The account for this difference leads to metric terms that include the derivatives of unit directional vectors.

Other options for cell-based filtering of velocities will rely on a much larger stencil. Based on triangles $c'$, $c''$ and $c'''$ (see Fig. 1 (c)) one can estimate $\nabla\overline{\mathbf{u}}_\ell$ on triangle $c$. Combining such estimates on $c$ and $c'$, the gradient will be estimated on edge $vv''$ and similarly on other edges, and then the divergence of such gradients will be computed at $c$. On uniform meshes this will involve a stencil of 10 triangles. Yet another method is to use the vector invariant form $\Delta = \nabla\nabla\cdot - \mathrm{curl\,curl}$. For centroidal velocities both the discrete divergence and vorticity are naturally computed on vertices, through the cycle over the boundary of median-dual control volumes. The operation of gradient and second curl will involve three vertex values and return the result to cells. In this case the stencil will occupy 13 triangles on uniform meshes. In can be shown that such Laplacians are not more accurate than the vertex-based Laplacian, but will lead to more expensive matrix vector multiplications in the solver procedure. We therefore did not try these Laplacians thus far.

For the C-grid type of discretization of MPAS-Ocean and ICON-o the vector invariant form of Laplacians presents the main interest. The locations of normal velocities is given by small red circles in Fig. 1 (b) and (d). The divergence will be computed at vertices for the hex-C grid and at triangles for triangular C-grid, and vice versa for relative vorticies. On uniform meshes 11 normal velocities will be involved in computations. These operators are not implemented yet, but this might be done in future.

## 2.2 Implementation

The computational process is divided into two phases: initialisation and solution. The initialisation stage is independent of the filtering scale and involves precomputing auxiliary arrays that are reused during the solution phase. This optimization strategy significantly reduces the overall computational time, particularly for large-scale problems. Python 3 was chosen as the primary programming language due to its widespread adoption in the Earth Sciences community and its extensive ecosystem of libraries. The implementation is publicly available as open-source software on GitHub.





### 2.2.1 Initialisation

The implementation of the implicit filter method involves precomputing several auxiliary arrays based on the mesh connectivity matrix. These arrays are independent of the filtering scale and can be reused for multiple filter applications. The non-zero co-
efficients of the D operator are determined using equations (3) and can be efficiently computed using sparse matrix algebra. To further enhance the computational performance, JAX's just-in-time compilation and vectorization capabilities were employed, resulting in a speedup of approximately 100x compared to a pure NumPy implementation. The precomputed auxiliary arrays can be saved to disk to reduce the computational overhead for subsequent filter applications. This approach leverages NumPy's file I/O capabilities and promotes efficient reuse of precomputed data across multiple filter scales.

### 2.2.2 Solving linear system

After computing the coefficients of the D operator, they are assembled into a sparse matrix using SciPy's implementation of compressed sparse column (CSC) matrices. The S operator is then calculated based on either equation 4 or 5. Following the suggestion of Guedot et al. (2015), the product of S and $\phi$ is subtracted from $\phi$ to simplify the solution of the linear system (1). The resulting modified $\phi$ is then used by the conjugate gradient solver along with S. A solver tolerance of $10^{-9}$ was used for
convergence.

Several alternative solvers, including the generalized minimal residual method (GMRES), were tested, but the conjugate gradient method consistently exhibited the best performance. The use of a preconditioner (incomplete LU factorization) was also investigated, but it did not lead to significant performance improvements.

To harness the parallel processing capabilities offered by GPUs, the CuPy library was employed. CuPy provides an identical
interface to SciPy and requires minimal modifications to the method implementation. This library is optimized for NVIDIA GPUs but also supports graphics cards from other vendors, such as AMD. By utilizing CuPy, the algorithm maintains independence from both the operating system and hardware vendors.

### 2.3 Box filter

To make a comparison with the explicit filter, it is necessary to implement the box filter method proposed by Aluie et al. (2018).
The crucial aspect involves defining the convolution kernel. Following formula was used:

$$G(\mathbf{x}) = A(1 - \tanh(c(|\mathbf{x}| - 1.75/k_\ell)/a)) \tag{8}$$

Here $A$ is a normalising factor, ensuring that $\int G d\mathbf{x} = 1$, $c$ is the filter sharpness factor set to $1.5$ and $a$ is a mesh resolution.

To implement this method, JAX was used to create the kernel matrix, while SciPy (in case of CPU) and CuPy (in case of GPU) were employed to apply the matrix to the data.



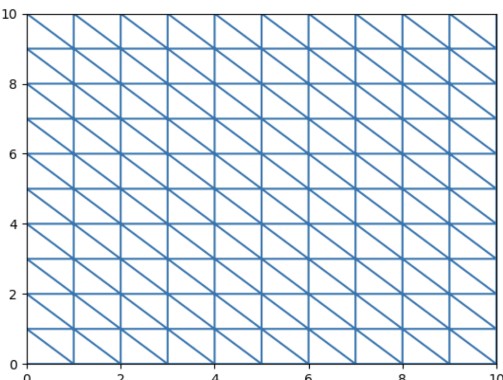

**Figure 2.** A regular rectangular mesh is converted into a triangular mesh by bisecting each rectangle into two triangles.

## 3  Data

The model data used was generated by the Finite-Element/volumE Sea ice-Ocean Model version 2 (FESOM2) simulation. FESOM2 is a multi-resolution sea ice-ocean model that solves the ocean primitive equations on unstructured meshes (Danilov et al. (2017)). The sea ice module, that is part of the model, is formulated on the same meshes as the ocean module. Configuration of the model has horizontal resolution of 1 km in the entire Arctic ocean which smoothly coarsens to 30 km in the rest of the global ocean. There are 70 $z$-levels in the vertical direction, with 5-meter spacing within the upper 100 meters. ERA5 atmospheric reanalysis fields (Hersbach et al. (2020)) were used to force the model. The model was initialized from the PHC3 climatology (Steele et al. (2001)) and run for 11 years starting from 2010. The first four years were considered as a spinup. The realizations of velocity field from the last seven years (2014–2020) were used in this work.

In order to be able to directly use the box filter on this dataset it was interpolated to a regular rectangular grid with 0.01 degree resolution. Prior to interpolation, the coordinate transform has been performed such that the Arctic Ocean corresponds to the equatorial region in new coordinates. Using this rotation minimizes the error caused by using a regular longitude/latitude grid, since it ensures that the grid cells are very close to squares. Linear interpolation was used. As this is a very costly process, the domain was restricted to areas north of 73 degrees latitude on the original mesh. The final mesh had dimension of 3200 by 3200 cells, resulting in 10.240.000 cell mesh.

Since the present implementation only supports triangular meshes (the support to other mesh types will be added in future), to apply both convolution-based and implicit filters, a new triangular mesh was constructed based on the previously mentioned regular mesh. This triangular mesh is obtained by splitting each cell of the regular grid into two triangles as shown in Fig. 2.





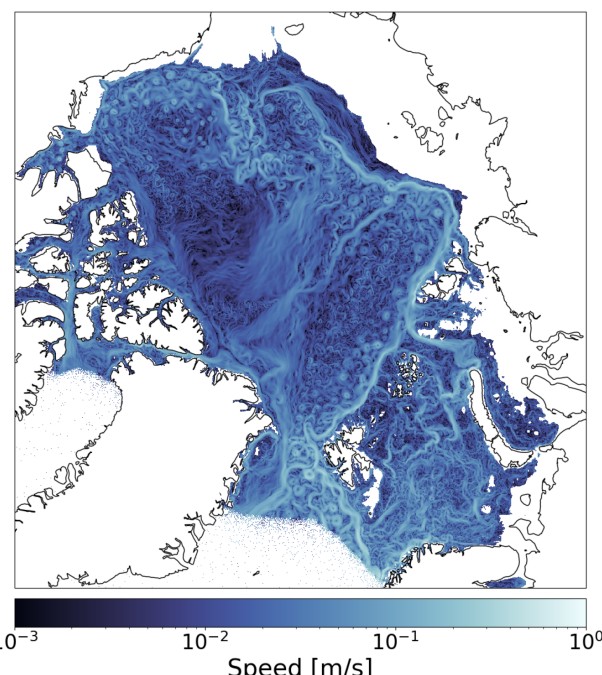 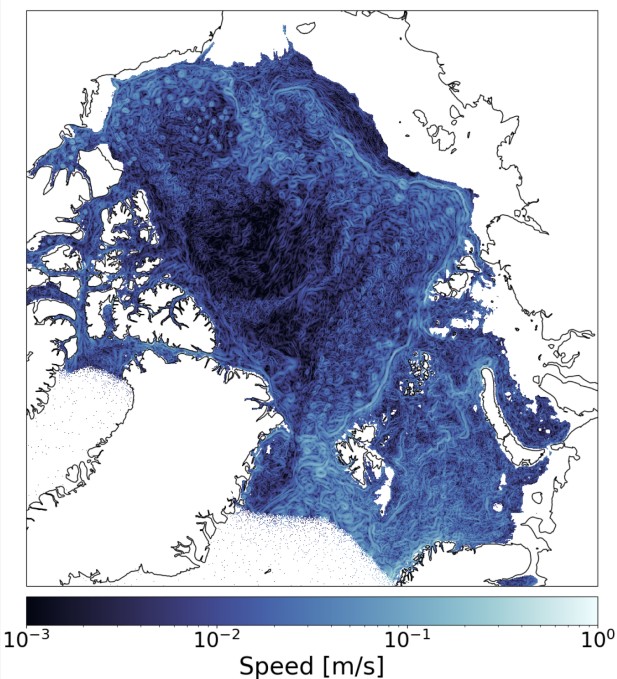

**Figure 3.** A snapshot of ocean currents at 70 m depth (logarithmic scale). The left panel shows the simulated results, while the right panel shows the results of high-pass filtering with the scale of 100 km. The implicit Laplacian filter has been used.

## 4 Results

Figure 3 illustrates the application of the implicit filter for high-pass filtering of the velocity field. The left panel shows a
snapshot of the absolute value of velocity at 70 m depth. Together with eddies and jet flows it shows a region with smoothed velocities, presumably caused by the sea-ice drift. The velocity field in the right panel is obtained by subtracting the velocity field coarse grained with the scale of 100 km, which leaves only the small scales. Using the implicit filter allows one to perform this operation on the native mesh and on the spherical Earth surface, without the need of regridding the data. The continental-break currents are rather strong and carry a significant part of kinetic energy. As is seen comparing the panels of Fig. 3, they
contribute very substantially into large-scale part of the flow. One can expect therefore that energy spectra computed for the entire Arctic Ocean will have an elevated spectral density at large scales.

Figure 4 presents kinetic energy spectra obtained with the implicit Laplacian filter using the original data (on the original triangular mesh) (red line) and interpolated data (blue). They are compared with the spectrum obtained by coarse-graining with the explicit box filter (green). The later is computed ignoring the Earth curvature (the cosine of latitude is replaced with 1),
enabling the convolution via the Fourier transform. The implicit filter allows us to compute the spectra of the interpolated data on both the longitude-latitude mesh and on its flat geometry approximation. Although the coarse-grained velocities show some



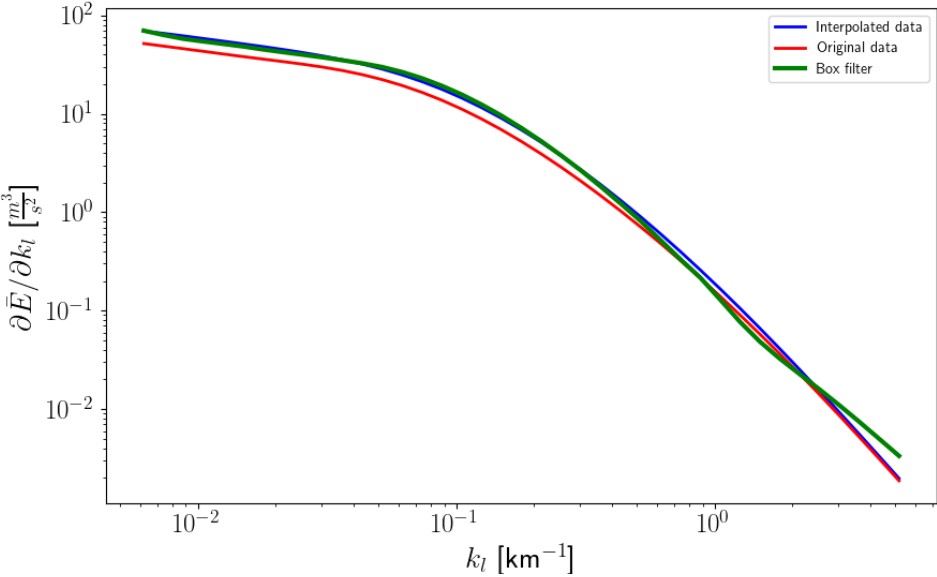

**Figure 4.** Wavenumber spectra for entire Arctic Ocean. Red line corresponds to the implicit harmonic filter applied to the data on the original grid. The blue and green lines correspond to the implicit harmonic and explicit box filter applied to the interpolated data. The highest wavenumber is $\pi/h$, where $h$ is the height of triangles of the original grid.

small differences in these two cases, the energy spectra turn out to be almost identical, which substantiates the approach taken for the box filter.

As seen in Fig 4, the implicit Laplacian filter and the explicit box filter provide matching results. The largest wavenumber

is $\pi/h \approx= 4$ cycle/km where $h$ is the height of triangles of the original mesh. One can note that for larger scales there is a small shift between the results computed on original and regular meshes. We guess that this is related to the effect of no-flux boundary conditions (Danilov et al. (2023)) which are applied along the boundary between water and land on the original mesh, but along the boundary of interpolation mesh in the other case. Interestingly, for the interpolated data, the spectra for the implicit and box filter are very close despite the difference in boundary conditions. Note that the spectra obtained using the

implicit Laplacian filter are smoother than spectra based on the box filter. This correlates with better spectral sensitivity of the latter, which is, however, worse than the sensitivity of the biharmonic filter.

In Fig. 5 we compare the spectra computed with harmonic and biharmonic filters. The convergence of biharmonic filters is much slower and can even be lost on large scales, so we stopped on the wavenumber $k_\ell \approx 0.025$ cycle/km, which corresponds to the wavelength of 250 km. While the spectra are close on the wavelength larger than approximately 12 km, they start

to deviate at smaller wavelengths. This could have numerical explanation at wavelengths that correspond to grid scales, as discrete Laplacians deviate from their continuous counterpart at grid scales, and this deviation is larger in biharmonic operators.




However, the spectra in Fig. 5 disagree also at larger scales, which is an indication that the real spectra have a slope steeper than $-3$ at these scales. We give elementary illustrations in section 6.

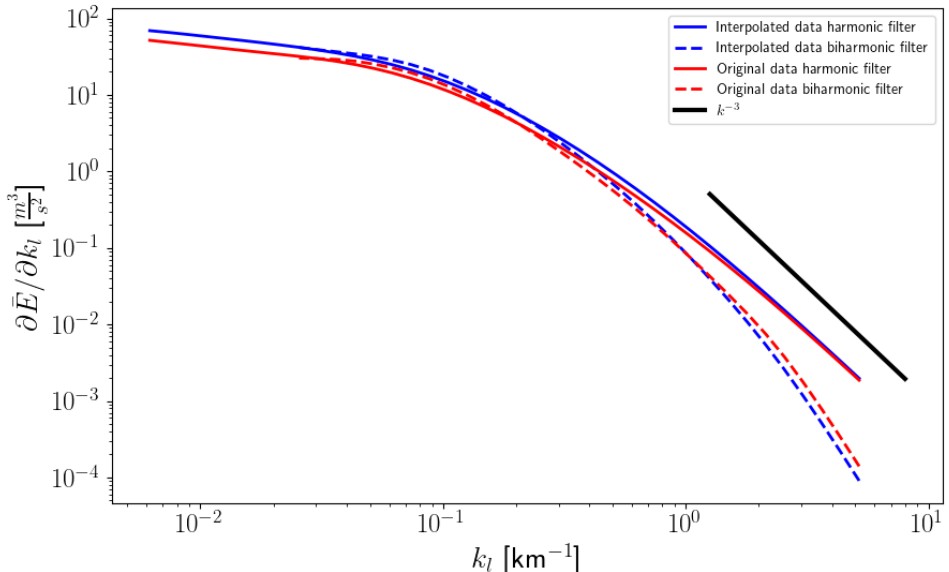

**Figure 5.** Comparison between the spectra computed with the box filter and the implicit harmonic filter applied to data on native grid and interpolated to regular grid.

## 5 Performance benchmarks

To facilitate the computational demands of this study, extensive computational resources were generously provided by the Jülich Supercomputing Centre (JSC). As the primary computing environment, a single node from the JUWELS Booster Module was exclusively utilised. This high-performance node boasts an impressive configuration, featuring two AMD EPYC Rome 7402 CPUs, 512 GB of DDR4 RAM, and four NVIDIA A100 GPUs, each equipped with 40 GB of HBM2e memory. To optimise computational efficiency and resource utilisation, only a single GPU was employed for the duration of this study.

The performance evaluation of the implicit filter method focuses on its computational efficiency, assessed by measuring the execution time required to process data. However, data access from disk can introduce substantial overhead, potentially influencing the overall execution time. To isolate the computational efficiency of the method, IO time, representing the duration spent reading and writing data to disk, is excluded from the execution time measurements. To ensure the accuracy and consistency of time measurements, each configuration is executed ten times, and the mean value is taken.

As evident from Figure 6, the performance of the implicit filter method exhibits a linear relationship with mesh size above $10^6$ nodes. Furthermore, the method effectively handles meshes with over 11 million nodes, achieving processing times of



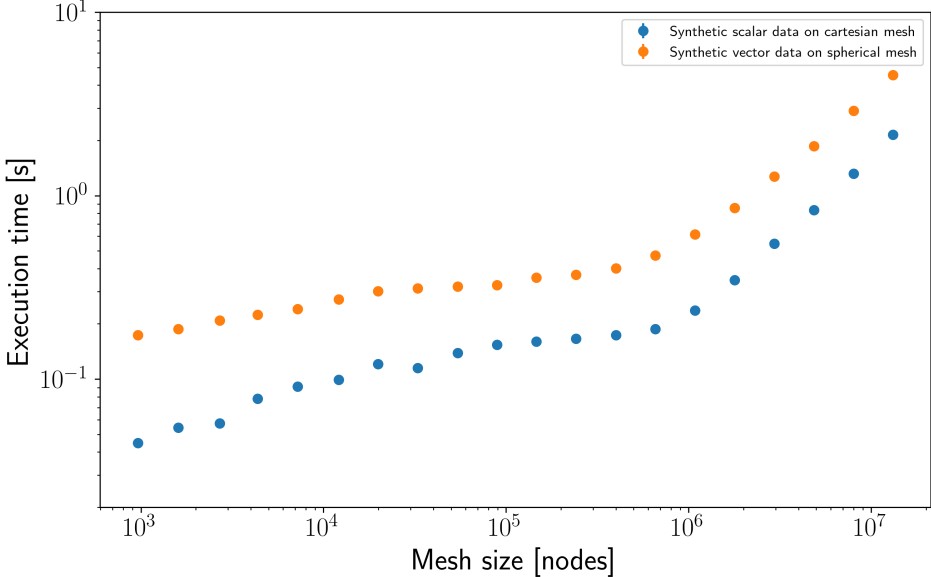

**Figure 6.** Execution time of implicit filter on synthetic data. Presenting both data on a Cartesian and spherical mesh. Filter size of 100 km was used at all cases.

approximately 5 seconds for a 100-kilometer filter scale. Such remarkable efficiency is attributed to the method's inherent scalability, enabling it to process data efficiently on increasingly large meshes without performance degradation.

This capability to handle large meshes is essential for analyzing real-world datasets, which often cover vast areas and demand high-resolution meshes for accurate representation. The implicit filter's scalability ensures its effectiveness in processing these large datasets, making it suitable not only for current state-of-the-art meshes but also for future generations of increasingly high-resolution meshes.

The measured results of the execution time, as shown in Fig. 7, exhibit a close-to-linear dependency with filter scale for those larger than 50 km. In particular, computation for a filter scale up to approximately 100 km is done within 6 s, even for a mesh with more than 10 million nodes and resolution about 1 km in the focus area. As this is the range of scales that is of the most interest, it shows remarkable performance. Convergence becomes more challenging for larger scales, and results diverge from a linear dependency.

## 6 Some comments

The present implementation supports triangular meshes of FESOM2 and produces coarse-grained fields at mesh vertices. Since the original discrete horizontal velocities in FESOM2 are at the centers of triangles, computation of coarse-grained velocities at triangles was also tried and found to lead to a very close results as concerns kinetic energy spectra. Vertex computations

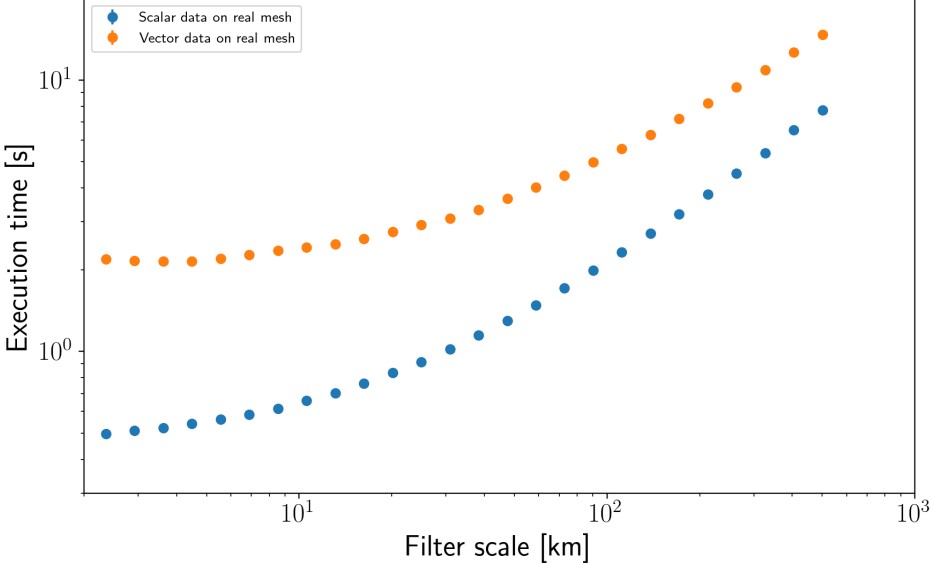

**Figure 7.** Execution time of implicit filter on FESOM2 output. Results show computation time of both vector and scalar data on 11 M node mesh (see Sec. 3).

should therefore be preferred, as they require smaller matrices. Nevertheless, triangle-based computations might be required if dissipation is studied, as the dissipation tendency is noisy. They are under implementation and are based on the simplified Laplacian (7). The support of C-grid type discretization for ICON-o discretization (scalar Laplacians for data on circumcenters

of triangles and vector-invariant Laplacians for velocities normal to edges) and the support of regular quadrilateral C-grids will be added in the nearest future. Our aim is a tool supporting different meshes.

We note that despite both the Laplacian and box-type filters are of the same order, the box-type filter has a better spectral resolution, as can be seen from the comparison of form factors in Danilov et al. (2023). For smooth and sufficiently flat energy spectra the difference between the spectra computed by both methods is minor, as in the case shown in Fig. 4.

However, the Laplacian and box-type filters cannot distinguish slopes steeper than $-3$, and higher-order filters are needed in this case. In eddy resolving numerical simulations energy spectra commonly steepen at grid scales as the consequence of dissipation, and this behavior can be masked if the Laplacian or box-type filters are used, as is the case with the data analyzed here. In order to further illustrate this point, in Fig. 8 we present the spectra computed using the implicit filters of different order given the Fourier spectrum $E_k \sim k^{-2}/(1 + (k/k^*)^2)$, with $k^* = 30k_{min}$, where $k_{min}$ corresponds to the domain wavelength.

The spectra were computed using the analytical expression for the form factor and the Fourier symbol of one-dimensional discrete Laplacian given in Danilov et al. (2023). While one expects that the Laplacian filter will fail for large $k$ in this case, the thick red line has a slope flatter than $-3$ over a range of wavenumbers where the real slope is $-4$. One can erroneously interpret this interval as a true spectrum (since it is flatter than $-3$). This behavior is caused by aliasing from the side of small





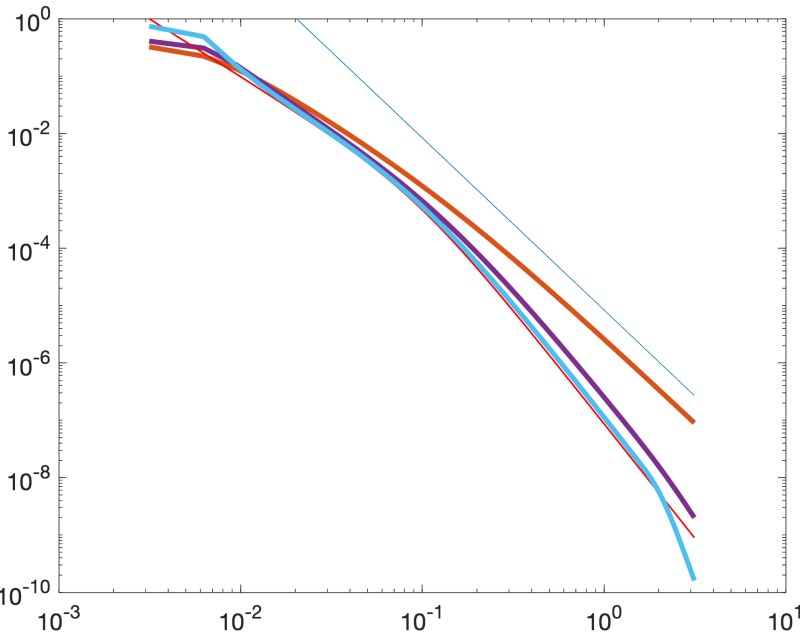

**Figure 8.** Theoretical energy spectra obtained using Laplacian (thick red), bi-Laplacian (violet) and four-Laplacian implicit filters. The Fourier spectrum is shown by thin red line and the straight line corresponds to a slope of $-3$.

wavenumbers where the spectral energy density is much larger. In contrast, the biharmonic filter (violet line) returns a spectrum

that is much closer to the Fourier spectrum, and the four-harmonic filter (thick blue line) is even closer. The drop-off at grid scales seen for the blue line is due to errors of the discrete Laplacian.

    The illustration in Fig. 8 stresses the fact that one generally needs to test the results obtained with the second-order filters if they show spectral slopes approaching $-3$ by comparing with the results of higher-order filters. Guided by this fact, the behavior shown in Fig. 5, and the possibility to compute the Fourier spectrum for the interpolated data, we compare the spectra

obtained with implicit harmonic and biharmonic filters with the Fourier spectrum in Fig. 9. The spectrum obtained with the harmonic filter deviates from the Fourier spectrum, but the biharmonic filter follows the Fourier spectrum very closely.

    The use of the biharmonic filter is computationally more expensive, and even worse, the convergence can be lost for large $\ell$ which correspond to wavelengths of domain size in the case of very fine meshes if the conjugate gradient solver is used. At present, we rely on the conjugate gradient solver available in python, and we work on preconditioners and solution methods

that will remove these difficulties. The improved convergence for biharmonic filter is important, as it opens up perspectives of using filters of higher order, as explained in Guedot et al. (2015) and Danilov et al. (2023).

    It is reminded that the wavenumber scale $k_\ell = 1/\ell$ used by us corresponds to the wavenumber of the Fourier spectrum. This means that $\ell$ corresponds to $\lambda/(2\pi)$, with $\lambda$ the wavelength, and that $\ell_{\text{box}} \approx 3.5\ell$.




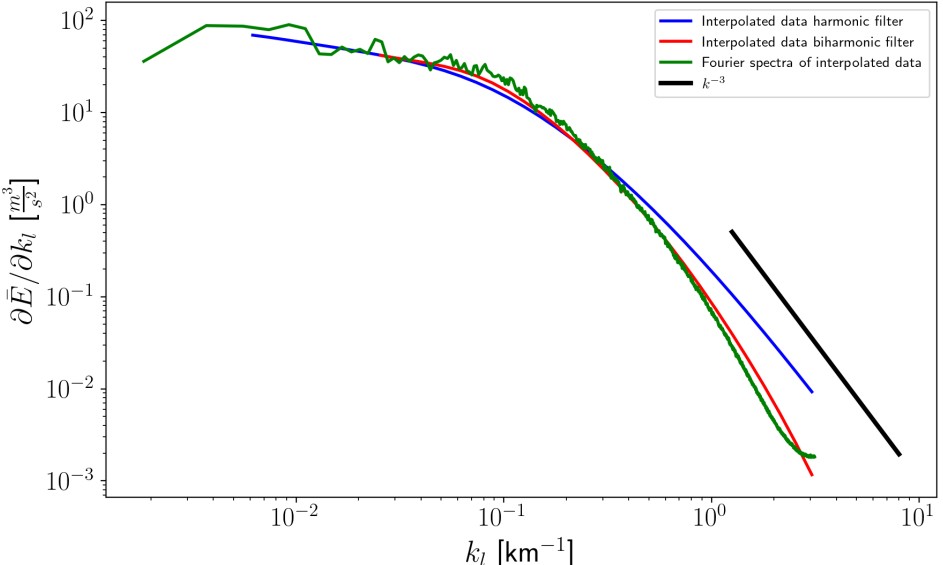

**Figure 9.** Fourier energy spectrum compared to the spectra obtained with implicit harmonic and biharmonic filters for the data interpolated to a regular grid.

## 7 Conclusions

This work presents a high performance implementation of a novel method for extracting spatial spectra from unstructured mesh data, offering a compelling alternative to conventional methods. The open-access code and elementary documentation can be found at GitHub (https://github.com/FESOM/implicit_filter) and Zenodo (Nowak and Danilov (2024)). Unlike its predecessors, the implicit filter method directly operates on unstructured meshes, such as triangular and quasi-hexagonal meshes, eliminating the need for computationally expensive interpolation to regular grids. This capability makes the implicit filter directly applicable to the output of unstructured-mesh ocean circulation models, surpassing the limitations of traditional methods.

To enhance practical applicability, the implicit filter method is implemented in Python using a high-performance algorithm that employs a two-phase approach to optimize computational efficiency. The first phase involves precomputing mesh-specific data, significantly reducing the computational load during the actual filtering process. This optimization strategy ensures efficient resource utilization and minimizes overall execution time. The second phase leverages cutting-edge sparse matrix algebra and GPU acceleration, harnessing the power of modern graphics processing units to achieve unparalleled performance and scalability. This computational prowess enables the processing of high-resolution data from meshes with millions of surface vertices within seconds.

The efficacy of the implicit filter method is demonstrated by applying it to compute spatial spectra of ocean currents from high-resolution General Circulation Model output. The results obtained from the proposed method exhibit agreement with



those obtained using traditional methods, such as box filter, validating its accuracy and robustness. Furthermore, the method's ability to handle unstructured meshes directly provides a more comprehensive analysis compared to traditional methods that require interpolation to a regular grid. However one needs to note that for spectra with slopes steeper than $-3$ it is needed to use biharmonic filter.

*Data availability.* All scripts used for making figures presented in this work are available on Zenodo. (Nowak et al. (2024)).

Data used for Figure 3 is not included as it would require files with size exceeding capacity of Zenodo repository. As this figure is only for illustrative purposes, similar figure can be obtained using any data and included example scripts. (Nowak and Danilov (2024))

*Code and data availability.* Source code along documentation and examples are available at: https://github.com/FESOM/implicit_filter and Zenodo (Nowak and Danilov (2024))

*Author contributions.* KN: software development, manuscript writing, data analysis, SD: manuscript writing, supervision, VM: running

simulation, software testing, CL: processing data, software development

*Competing interests.* The authors declare that the research was conducted in the absence of any commercial or financial relationships that could be construed as a potential conflict of interest.

*Acknowledgements.* This publication is part of the EERIE project funded by the European Union. Views and opinions expressed are however those of the author(s) only and do not necessarily reflect those of the European Union or the European Climate Infrastructure and Environment

Executive Agency (CINEA). Neither the European Union nor the granting authority can be held responsible for them.

This work has received funding from the Swiss State Secretariat for Education, Research and Innovation (SERI) under contract 22.00366. This work was funded by UK Research and Innovation (UKRI) under the UK government's Horizon Europe funding guarantee (grant number 10057890, 10049639, 10040510, 10040984).

This work is also a contribution to projects M3 and S2 of the Collaborative Research Centre TRR181 "Energy Transfer in Atmosphere

and Ocean" funded by the Deutsche Forschungsgemeinschaft (DFG, German Research Foundation) - Projektnummer 274762653.

Authors acknowledge usage of ChatGPT by OpenAI version 3.5 in preparation of the manuscript. It was used for linguistic and editorial purposes.



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
