# Peer review of "Implementation of implicit filter for spatial spectra extraction"

_EGUsphere, 2024_

## Author Response (AR1)

**Reply to the reviewer's comments**

We thank both reviewers for their constructive feedback and thoughtful suggestions, which have helped us improve the clarity, rigor, and completeness of our manuscript. Below, we provide a detailed response to the comments raised by Reviewer 1 (RC1) and Reviewer 2 (RC2), along with an outline of the revisions made to the manuscript.

Reviewer 1:

1) Some of the language is overblown and sounds more like a sales pitch than a scientific paper
   We appreciate the feedback on the language and have revised the manuscript to adopt a more neutral and scientific tone.

2) There is a lot of content describing how to solve a Poisson equation on an unstructured mesh. These are established techniques and so perhaps should be moved to an appendix?
   We believe that not every reader of the article must be familiar with those techniques. Thus, in our opinion it's better for the article main part to contain that information, as they are necessary for the understanding of its contents

3) Please describe more clearly how you calculate the wavenumber spectra for the original data, for the box filter and for the interpolated data.
   We have expanded the explanation of how wavenumber spectra are computed for the original data, box filter, and interpolated data. Additionally, we updated it, taking into account results presented by Zhao & Aluie, which was suggested by reviewer 2.

4) Please define what you mean by "convergence of biharmonic filters". In what way are these iterative and what are you trying to converge towards?
   We have clarified the term "convergence of biharmonic filters" in the manuscript.

5) My understanding is that you have smoothed the data on the native grid by solving a Poisson equation. The solution of a Poisson equation on an unstructured grid is established. I thought that you still have to interpolate onto a lat-lon grid to calculate power spectra. You are just interpolating coarser data. Please explain.
   The interpolation to the lat-lon grid is not required at any step of obtaining the power spectra. The power spectra can be computed directly on unstructured mesh, simply by computed in the same way as on regular grid, however values need to weighted by size of a mesh element they are assigned to. In case of regular mesh this step is unnecessary as all elements have the size.

Reviewer 2:

Substance:

1) In section 4 the differences in the spectra shown in Figures 4 and 5 as 'small' (l197) and the spectra are described as 'matching' (l199). These differences may appear small on a log-log plot, but when I try to estimate by hand I find that the difference in KE at large scales could be as much as 50%. Please compute the percent change so that we can know how small it actually is.

We calculated the percent differences in kinetic energy at large scales to provide a quantitative assessment of the discrepancies. This replaces the qualitative descriptions such as "small" or "matching."

2) The manuscript mentions in a few places convergence problems for the biharmonic filter with large filter scale. Preconditioning is mentioned as a possible way to alleviate this in future versions of the software. The original implementation of gcm-filters (Loose et al., 2022) also suffered from difficulties with large filter scales, as described by Grooms et al. (2021). We originally suggested coarsening before applying the filter with a large filter scale, which might also work here. Ultimately we corrected the numerical stability problem by changing the way the explicit filter is applied to data, with the stable method available in versions 0.3 and later, as described here
We plan to address the issue with convergence by developing custom preconditioning technique, specific to this method.

3) The scaling in figures 6 and 7 is described as 'linear.' To my eye the data on the log-log plot are plausibly linear in some range, which implies power-law scaling, but it's not clear to me that the slope is 1, which would imply linear scaling. It would be beneficial to estimate and report the exponent of the power law over the ranges where linear scaling is claimed.
We estimated and reported the exponent of the power-law scaling in the relevant range of the log-log plots and added fitted linear function to the plots.

4) The filter has two stages, setup and online. While the online performance is impressive, the setup cost is never recorded. Presumably the setup cost is not prohibitive, but it would be beneficial to report setup times.
We added the setup times for the filter in Section 4, alongside the online performance results.

Clarity:

1) The term 'box filter' is used here to refer to a convolution-type filter. Elsewhere in the literature the term 'box filter' is sometimes used to refer to a convolution-type filter where the kernel is an indicator function, often on a rectangle. The kernel used here for the 'box filter' is smooth and only approximately has compact support. The

authors might consider changing the term 'box' to avoid giving an incorrect impression that their kernel is an indicator function on a rectangle.
For the sake of clarity, we changed the name from "box filter" to "convolution filter".

2) Figure 2 illustrates a regular grid. Although the aspect ratio of the data is one, the aspect ratio of the plot is not one. The authors might consider forcing the aspect ratio of the plot to be one.
We don't see an issue with the format, as this image is only illustrating mesh structure.

3) In section 4 some of the filters use data on dry/land cells. Are the values on dry cells are set to zero?
Yes, they are assumed as 0

4) Lines 195-196: "The implicit filter allows us to compute the spectra of the interpolated data on both the longitude-latitude mesh and on its flat geometry approximation." This sentence is clear, but it's not clear to me which plots use the interpolated data in spherical geometry and which plots use the interpolated data in flat geometry.
We added explicit information that in figure 3, data with spherical geometry was used.

5) Several places mention meshes with over 11 million nodes, but the high resolution mesh is earlier mentioned to have 10.24 million nodes. Please clarify.
AO_40 mesh that was used during benchmarking has 11538465 surface nodes. This information was added to the description of the benchmark.

6) The paragraph on lines 233-237 says that scaling is linear for filter scales larger than 50 km, and it also says that the results diverge from a linear dependency for filter scales larger than 100 km. This is inconsistent. See also substance comment #3.
We revised the paragraph to make it clear regarding linear scaling for filter scales larger than 50 km and when convergence issues can appear.

7) The discussion in section 6 about extracting spectra with slopes steeper than -3 is heavily indebted to previous work on the subject, including Sadek & Aluie (2018). Although this paper is cited in the introduction, it is not cited in section 6; I think it would be appropriate to acknowledge the provenance of these ideas locally in section 6. The authors might also be interested in the preprint recently posted by Zhao & Aluie on the same topic
Included new results presented by Zhao & Aluie (2024), which significantly improved performance of method presented here.

8) We fixed mentioned typos. We appreciate the throughout check of our article.

---

## Author Response (AR2)

**Reply to the editor's comments**

Dear Dr. Marti,

Thank you for your for the opportunity to provide these clarifications regarding our revised manuscript. We appreciate your careful review.

Regarding your points:

1. **Convergence of biharmonic filters:** We have now removed this term and explicitly provide the details about conjugate gradient iterations and convergence criteria. (lines 216-217)
2. **Kinetic energy spectra:** We have incorporated the calculated percent differences in kinetic energy at the largest scale into Section 4 (lines 205 and 207)
3. **Typos:** Thank you for pointing these out. The typographical errors have been corrected

We trust these modifications and clarifications address your concerns. We have attached the newly revised manuscript incorporating these changes.

Best regards,

Kacper Nowak

On behalf of all authors.